# Fully Aromatic Thermotropic Copolyesters Based on Vanillic, Hydroxybenzoic, and Hydroxybiphenylcarboxylic Acids

**DOI:** 10.3390/polym16111501

**Published:** 2024-05-25

**Authors:** Pavel A. Mikhaylov, Kirill V. Zuev, Yaroslav V. Golubev, Valery G. Kulichikhin

**Affiliations:** A. V. Topchiev Institute of Petrochemical Synthesis, Russian Academy of Sciences (TIPS RAS), 29 Leninsky Prospekt, 119991 Moscow, Russia; ya_golubev@ips.ac.ru (Y.V.G.); klch@ips.ac.ru (V.G.K.)

**Keywords:** aromatic polyester, thermotropic liquid crystal polymer, TLCP, polycondensation, polymer analysis, DSC, TGA, bio-monomer, vanillic acid

## Abstract

Several series of new polymers were synthesized in this study: binary copolyesters of vanillic (VA) and 4′-hydroxybiphenyl-4-carboxylic (HBCA) acids, as well as ternary copolyesters additionally containing 4-hydroxybenzoic acid (HBA) and obtained via three different ways (in solution, in melt, and in solid state). The high values of logarithmic intrinsic viscosities and the insolubility of several samples proved their high molecular weights. It was found that the use of vanillic acid leads to the production of copolyesters with a relatively high glass transition temperature (~130 °C). Thermogravimetric analysis revealed that the onset of weight loss temperatures of ternary copolyesters occurred at 330–350 °C, and the temperature of 5% mass loss was in the range of 390–410 °C. Two-stage thermal destruction was observed for all aromatic copolyesters of vanillic acid: decomposition began with VA units at 420–480 °C, and then the decomposition of more heat-resistant units took place above 520 °C. The copolyesters were thermotropic and exhibited a typical nematic type of liquid crystalline order. The mechanical characteristics of the copolyesters were similar to those of semi-aromatic copolyesters, but they were much lower than the typical values for fully aromatic thermotropic polymers. Thus, vanillic acid is a mesogenic monomer suitable for the synthesis of thermotropic fully aromatic and semi-aromatic copolyesters, but the processing temperature must not exceed 280 °C.

## 1. Introduction

Thermotropic main-chain polyesters (TMCPs) are a class of high-performance polymers with a number of unique properties [1,2,3,4,5,6,7,8,9]. TMCPs are capable of forming highly ordered liquid crystalline melts and do not require solvents during processing, unlike materials produced from lyotropic solutions such as aramid fibers. The TMCP backbone contains aromatic rigid units such as 1,4-phenylene, 2,6-naphthalene, and 4,4′-biphenylene, enabling the formation of highly ordered nematic melts. This high orientation of macromolecules in the melt allows for the production of fibers and molded articles with high tensile modulus and tensile strength. Co-extrusion of TMCPs and conventional thermoplastics results in self-reinforced composites or in situ composites [9,10,11,12,13,14]. The mechanical properties of TMCP fibers are comparable to those of aramid fibers [6,15,16]. For instance, the tensile strength of Vectran HS fibers was found to be ~3 GPa [15]. An early patent showed that the tensile strength of Kevlar fibers was only slightly higher than that of TMCP fibers (36.9 g per denier vs. 30.8 g per denier) [16]. It should be noted that the moisture uptake of TMCPs is significantly lower than that of aramid fibers [3]. Typically, TMCPs possess high chemical and heat resistance, along with good thermal stability and a low thermal expansion coefficient. Thus, TMCPs are used in the aerospace field, as well as in electronic or optoelectronic devices. Due to their high thermal and heat resistance, TMCPs can be used to produce ovenware [17].

Generally, TMCPs represent copolyesters consisting of several monomeric units, because homopolymers composed from rigid units such as 4-hydroxybenzoate do not melt without decomposition [3,18]. Usually, the TMCP backbone contains rigid mesogenic aromatic para-substituted units with a bond angle of 180 °C, aromatic units with a bond angle different from 180 °C (such as 1,3-phenylene so-called “kinked units”), aromatic units with bulky substituents such as phenyl terephthalic acid and chlorine hydroquinone, and a flexible aliphatic spacer such as ethylene glycol. Copolymerization of two or more monomers, introducing “kinked” units and/or aliphatic spacers, enables reducing the melting point and obtaining melt-processable products [1,8]. TMCPs consisting of aromatic units without aliphatic flexible fragments in the main chain are known as thermotropic fully aromatic copolyesters, which possess exceptional chemical thermal stability (up to 500 °C in an inert atmosphere), unlike the TMCPs with aliphatic flexible spacers, which are known as aliphatic–aromatic or semi-aromatic TMCPs.

In short, fully aromatic TMCPs are produced from conventional oil-based monomers: hydroquinone; 4,4′-dihydroxybiphenol; 4-hydroxybenzoic, 6-hydroxy-2-naphthoic, terephthalic, isophthalic, and 2,6-naphthalenedicarboxylic acids; etc. Aliphatic–aromatic TMCPs are prepared through chemical modification of poly(ethylene terephthalate) by 4-acetoxybenzoic acid via ester–ester exchange (X7G Eastman Kodak, Rodrun Unitika), resulting in a copolymer of 4-hydroxybenzoic acid with flexible spacer units of PET [19].

The great interest in both the academic and industrial fields over the past decades has been the use of monomers from bio-based renewable sources. Lignin and cellulose are the most abundant renewable feedstocks. The most promising lignin-based polyester monomers are phenolic acids: vanillic, syringic, 4-hydroxybenzoic, floretic (3-(4-hydroxyphenyl)propionic), 4-hydroxycinnamic, ferulic (3-methoxy-4-hydroxycinnamic), dihydroferulic, etc. [20,21,22,23,24,25,26,27,28]. 4-Hydroxybenzoic and vanillic acids are aromatic self-condensable AB-type monomers. It should be noted that 4-hydroxybenzoic acid can be produced both from conventional oil-based sources and from renewable feedstocks, while the most promising source of VA is lignin. Structurally, vanillic acid is similar to 4-hydroxybenzoic acid, excluding the presence of the bulky side –OCH_3_ group, possessing mesogenic properties, and suitable as a comonomer for the synthesis of TMCPs. A role of the bulky side group of vanillic acid is that it disrupts the molecular order in copolymers, thereby reducing their melting or softening point. The majority of recent papers have focused on the investigation of semi-aromatic TMCPs based on vanillic acid.

In addition to their reduced thermal stability, there are several disadvantages of aliphatic–aromatic TMCPs. Aliphatic and aromatic monomers possess poor condensability, resulting in low molecular weights and very long polycondensation times [29,30,31,32]. The low thermal stability of aliphatic monomers prevents the use of high polycondensation temperatures.

In an early work, H. R. Kricheldorf investigated vanillic acid (VA) homopolymers [33,34] and some binary VA copolyesters and polyester amides were also described [34,35,36]. Polyvanillate was found to be an intractable polymer; however, some binary copolymers were melt-processable. Wilsens [37] described the possibility of using vanillic acid as a comonomer for fully aromatic copolyesters consisting of 4-hydroxybenzoic acid, 2,5-furandicarboxylic acid, hydroquinone, and 4,4′-diphenyldiol. However, Wilsens et al. did not obtain high molecular weights for fully aromatic copolyesters of vanillic acid, and they did not test the mechanical properties because they used thin-film polymerization of small amounts of monomers between glass slides.

In the present work, we synthesized and studied a series of binary copolyesters of vanillic and 4′-hydroxybiphenyl-4-carboxylic acid (HBCA). According to our previous studies, HBCA is a promising comonomer for TMCPs [19,38]. Unfortunately, the binary copolyesters did not demonstrate melting or softening at reasonable temperatures. Thus, a series of ternary copolyesters of vanillic, 4′-hydroxybiphenyl-4-carboxylic, and 4-hydroxybenzoic acids were obtained and studied. Some of the ternary copolyesters demonstrated melting or softening at ~300 °C; the melt-processable ones were converted into thin films by molding and then mechanically tested.

## 2. Materials and Methods

### 2.1. Materials

4-Hydroxybenzoic acid (HBA) and 3-hydroxybenzoic acid (3HBA) were commercial products with a purity of >98% and supplied by Acros Organics (India) and Reachem (Russia), respectively.

4-Acetoxybenzoic acid (4ABA) and 3-acetoxybenzoic acid (3ABA) were synthesized according to a route reported previously [33]: the corresponding acids were refluxed for 2 h in toluene with 20 mol% excess of acetic anhydride, and then the products were recrystallized from water. For 4ABA: ^1^H NMR (400 MHz, CDCl_3_) *δ*, ppm—8.18–8.12 (m, 2H), 7.26–7.18 (m, 2H), 2.34 (s, 3H). For 3ABA: ^1^H NMR (400 MHz, CDCl_3_) *δ*, ppm—11.68 (s, 1H), 8.00 (d, *J* = 7.8 Hz, 1H), 7.84 (d, *J* = 2.4 Hz, 1H), 7.50 (t, *J* = 7.9 Hz, 1H), 7.36 (dd, *J* = 8.0, 2.5 Hz, 1H), 2.34 (s, 3H).

4′-Acetoxybiphenyl-4-carboxylic acid (ABCA) was synthesized at Yaroslavl State Technical University (Russia). For ABCA: ^1^H NMR (400 MHz, CDCl_3_) *δ*, ppm—8.30 (t, *J* = 1.8 Hz, 1H), 8.10 (dt, *J* = 7.9, 1.4 Hz, 1H), 7.86 (dt, *J* = 8.0, 1.4 Hz, 1H), 7.68–7.62 (m, 2H), 7.58 (t, *J* = 7.8 Hz, 1H), 7.23–7.17 (m, 2H), 2.45 (s, 3H).

Food vanillin was used for the synthesis of *o*-acetyl vanillic acid after triple extraction with chloroform from aqueous solution using a separation funnel. The organic layer was dried over anhydrous Na_2_SO_4_, and the chloroform was evaporated under reduced pressure (melting point 84–85 °C).

Vanillic acid (VA) was prepared via the caustic fusion of vanillin, as previously described [39]. Briefly, KOH (150 g) and water (20 mL) were placed in a 2 L stainless steel vessel equipped with an overhead wire stirrer and heated to ~120 °C until a viscous solution was obtained. Vanillin (60 g) was added portion-wise to the vessel, and the temperature was increased to 170 °C for 30 min. The reaction mixture was cooled, recrystallized from water, and dried. The yield of VA was 80%, with a melting point of 207–210 °C.

*O*-acetyl vanillic acid (AVA) was prepared as previously described in [34]. Briefly, vanillin (21 g), acetic anhydride (20 mL), toluene (100 mL), and pyridine (1.25 mL) were refluxed for 3 h. The mixture was concentrated in vacuum, diluted with toluene, and concentrated again to remove acetic acid. A brownish residue was dissolved in toluene and then recrystallized via the portion-wise addition of petroleum ether. Additionally, it was recrystallized twice from chloroform via the portion-wise addition of petroleum ether. As a result, colored impurities were removed. The yield of AVA was 60%, with a melting point of 141–145 °C. For AVA: ^1^H NMR (400 MHz, chloroform-*d*) *δ*, ppm—7.76 (dd, *J* = 8.2, 1.9 Hz, 1H), 7.71 (d, *J* = 1.8 Hz, 1H), 7.14 (d, *J* = 8.2 Hz, 1H), 3.91 (s, 3H), 2.34 (s, 3H). ^13^C NMR (101 MHz, CDCl_3_) *δ*, ppm—171.28, 168.63, 151.30, 144.52, 128.03, 123.59, 123.11, 113.96, 56.23, 20.79. MRM ESI-MS, [M-H]^–^, *m*/*z*: 209.20 → 108.00; 209.20 → 123.00; 209.20 → 59.10.

NMR solvents (chloroform-*d*, DMSO-*d*_6_) with a purity of >99.8% were supplied by Cambridge Isotope Laboratories (USA). All auxiliary chemicals were supplied by Himmded (Russia) and Ekos-1 (Russia) and were used without further purification.

### 2.2. Polymer Synthesis

#### 2.2.1. Solution Polycondensation

Binary copolyesters of VA and HBCA (VA-HBCA-1/2, VA/HBCA-1/1, and VA/HBCA-2/1) were prepared via high-temperature solution polymerization in the inert solvent Thermolan H (NPK Polyester LLC, Russia), representing a mixture of hydrocarbons (bp > 330 °C) as previously described in [34]. Briefly, polycondensation of 10 mmol of a mixture of AVA and ABCA was carried out in a 100 mL three-neck round-bottomed flask equipped with an overhead stirrer, argon inlet, and short column. The mixture of monomers was immersed in the flask, and 50 mL of Thermolan (0.2 M concentration of monomers) was added. Argon was bubbled through the reaction mixture for 15 min, and then the flask was placed in a metal bath preheated to 300 °C. The flask was left in the metal bath for 16 h at 300 °C under a continuous flow of argon. Then, the flask was cooled under an inert atmosphere, and the product was collected on a filter and washed with hot acetone in a Soxhlet extractor for at least 8 h. The products are listed in Table 1.

#### 2.2.2. Small-Scale Melt Polycondensation

Small amounts of comonomer (AVA, 3/4ABA, ABCA) mixtures (~1 g) with different molar ratios were polymerized in 50-mL two-neck round-bottomed flasks connected with an argon/vacuum line. The flask was filled with argon heated to 240 °C and then gradually heated to 320 °C until the copolymer was obtained. The products are listed in Table 2.

#### 2.2.3. Melt Polycondensation

The melt polycondensation at 10–25 g scale was performed using the following procedure: A mixture of AVA, 4ABA, and ABCA was loaded into a 100 mL 3-neck round-bottomed flask equipped with an overhead mechanical stirrer, inert gas inlet, and vacuum outlet. The flask was evacuated and filled with nitrogen three times, and then it was immersed in a metal bath preheated to 240 °C under a slow flow of argon. After 60 min of stirring, the temperature was increased by 10 °C every 30 min until reaching 280 °C. After 30 min of stirring at 280 °C, the temperature was increased to 300 °C, and after another 30 min it was further increased to 320 °C. After 60 min of stirring at 320 °C, vacuum (<1 Torr) was applied for 30 min. The reaction mixture became too viscous for stirring, and it was removed from the flask under argon on polyimide film. The product was washed with hot acetone in a Soxhlet extractor for at least 8 h. The products are listed in Table 3.

#### 2.2.4. Melt Polycondensation Accompanied by Solid-State Polycondensation

The first step—polycondensation in the melt—was carried out similarly to the process described in Section 2.2.3, until the stage of heating the melt to 280 °C. After 30 min of stirring at 280 °C, the prepolymer was removed from the flask on polyimide film. The prepolymer was ground, placed in a vessel, and heated at 250–260 °C for 8–16 h under a continuous flow of argon or vacuum (<0.1 Torr). The products are listed in Table 4.

### 2.3. Comonomer and Polymer Characterization

The ^1^H NMR spectra of the comonomers were recorded on an AVANCE III HD spectrometer (Bruker, Germany) with an operating frequency of 400 MHz. DMSO-*d*_6_ or CDCl_3_ was used as the solvent.

The Shimadzu Series 20 liquid chromatograph with an LCMS 8040 liquid mass spectrometer (LC-ESI-MS/MS) was applied to analyze *o*-acetyl vanillic acid. A sample was dissolved in a 50:50 mixture of water and acetonitrile to a concentration of about 100 μg/mL. The mobile phase was composed of 0.1% HCOOH aqueous solution (50%) and 0.1% HCOOH solution in acetonitrile; the mobile phase flow rate was 0.5 mL/min. Electrospray negative ionization was used. The gas pressure in the collision cell was 230 kPa. The flow rates of drying and atomizing gases were 15 and 2.5 L/min, respectively.

The measurement of logarithmic viscosity ([*η*]) for the polymers was carried out at 60 ± 0.1 °C using an Oswald viscometer with a 0.6 mm capillary and calculated using Equation (1):(1)$$\left[\eta \right]=ln\left(t/t_{0}\right)/c$$
where *t* and *t*_0_ are the flow time of the polymer solution and of the pure solvent through the capillary, respectively, while *c* is the concentration of the polymer solution (0.1 g of polymer in 100 g of pentafluorophenol).

Fourier-transform infrared (FTIR) spectroscopy was performed using an IFS-66 v/s IR-Fourier spectrometer (Bruker, Billerica, MA, USA) in a wavelength range of 4000–600 cm^−1^. The IR spectra were obtained in attenuated total reflectance (ATR) mode.

The X-ray diffraction (XRD) patterns of the copolyesters were recorded using a Rotaflex RU-200 (Rigaku, Japan) device coupled with a rotating copper anode, a horizontal D/Max-RC goniometer, and a secondary graphite monochromator. The X-ray source mode was 50 kV–100 mA, and the region of diffraction angles (in 2*θ*) was 5–60 ° (scanning rate of 2 °/min with a step of 0.04°).

DSC/TGA thermograms of binary copolyesters were recorded with a TGA/DSC3+ system (Mettler Toledo, Columbus, OH, USA) in the following regime: heating from 30 to 1000 °C at a rate of 10 °C/min, with a 100 cm^3^/min inert gas (argon) flow rate.

DSC thermograms of ternary copolyesters were obtained using a DSC 204F1 instrument (Netzsch, Selb, Germany) in the following mode: heating from 20 to 350 °C at a rate of 10 °C/min, followed by cooling to 20 °C at a rate of 10 °C/min, and then reheating to 350 °C at a rate of 10 °C/min. The analysis was carried out in an inert atmosphere (argon—flow rate: 40 mL/cm^3^).

TGA analysis of ternary copolyesters was carried out using an STA 449F3 Jupiter instrument (Netzsch, Selb, Germany) in the following mode: heating from 25 to 1000 °C at a rate of 10 °C/min. The analysis was performed in an inert atmosphere (argon—flow rate: 50 mL/cm^3^).

The LC properties of the copolyesters were analyzed via polarizing optical microscopy (POM) on a 6 PO (Biomed, Moscow, Russia) device coupled with an FP900 thermal control system (Mettler, Columbus, OH, USA) and an E3ISPM5000 digital camera (ToupTek Photonics Co., Zhejiang, China). The temperature range of the measurements was 25–375 °C, at heating and cooling rates of 10–20 °C/min.

The mechanical characteristics of the ternary polyesters (tensile strength and relative elongation at break) were assessed during tensile testing of film samples (thickness 100 μm) on an I1140M-5-01-1 instrument (TOCHPRIBOR-KB, Ivanovo, Russia); a 100 N load cell was used, and the loading speed was 10 mm/min; the test temperature was 25 °C.

## 3. Results and Discussion

### 3.1. Binary Copolyesters

Unsubstituted and substituted derivatives of hydroxybenzoic acids are widely used for the synthesis of TMCPs [1,8]; however, 4-hydroxybenzoic acid itself is not quite suitable as a monomer because of the decarboxylation by-process at ~200 °C [18]. Therefore, acetoxybenzoic acids prepared via the acetylation of hydroxybenzoic acids are usually used as the monomers. The general route for synthesizing binary and ternary copolymers from acetoxy acids is presented in Figure 1.

There are three possible ways to synthesize polyesters: solution polycondensation (SP), melt polycondensation (MP), and solid-state polycondensation (SSP). High-melting-point polyesters, including non-meltable ones, can be synthesized via SP and SSP procedures. Copolyesters with melting or softening points lower than their decomposition point can be synthesized via the MP method. Thus, binary copolyesters of VA and HBCA (B1/2, B1/1, and B2/1) were synthesized by means of solution polycondensation from AVA and ABCA at 300 °C (Table 1). Thermolan H was used as a high-temperature solvent representing a mixture of aromatic hydrocarbons with a b.p. of >330 °C.

The binary copolymers were characterized by FTIR spectroscopy (Figure 2) because of their insolubility in common solvents. A very strong band of C=O vibration (*ν*_C=O_) of ester groups appeared at 1729 cm^−1^ for B1/2, shifted to 1731 cm^−1^ for B1/1 and to 1732 cm^−1^ for B2/1. This shift is entirely consistent with the fact that C=O vibration of the ester groups appeared at 1727 cm^−1^ for the HBCA homopolymer and at 1735 cm^−1^ for polyvanillate [34,40]. Benzene ring stretching vibrations (*ν*_C=C_) appeared around 1600 cm^−1^ and 1500 cm^−1^. Stretching vibration of C–O (*ν*_C-O_) bonds in VA units appeared at 1283 cm^−1^ and 1028 cm^−1^, and its intensity increased with the increase in the VA/HBCA ratio in the copolymers. Absorption peaks at 763 cm^−1^, 749 cm^−1^, 713 cm^−1^, and 694 cm^−1^ were related to the in-plane bending vibrations of para- and 1,2,4-substituted phenyl rings.

The thermal properties of binary copolyesters were analyzed via the DSC-TGA method. The glass transition points were slightly reduced from 135.5 °C to 132.5 °C with the increase in VA content in the copolyesters (Figure 3). The *T*_g_ points were higher than that of the copolyester Vectra A950 (*T*_g_ = 93 °C) composed of linear 2-hydroxy-6-naphthoic and *p*-hydroxybenzoic units [41]. This was explained by the effect of the bulky substituent –OCH_3_ in VA units, decreasing the chain mobility, which has been observed for other TMCPs [42]. DSC did not show melting of B1/2-B2/1, indicating that these polymers have a very low degree of crystallinity.

The TGA thermograms (Figure 4a) demonstrate that the decomposition onset temperature (*T*_on_) and the temperature of 5% weight loss (*T*_5%_) were reduced with the increase in VA content in the copolyesters. The DTG curves (Figure 4b) of the binary copolyesters show a two-stage decomposition process with two minima: the first one was observed at temperatures below 500 °C, and the second at temperatures higher than 500 °C. The first minimum belonged to decomposition of VA units, and the second to HBCA decomposition. Thus, increasing the molar fraction of VA units results in decreasing the thermal stability of copolyesters. Additionally, this is reflected in the amounts of mass remaining after decomposition when heating to 1000 °C: 30.1% for B1/2 and 23.9% for B2/1. Although B2/1 can be molded with short-term exposure at 350 °C, the TGA data indicate that such high temperatures are not applicable for B2/1, since material darkening was observed.

Copolyesters B1/2 and B1/1 were not processable because of their high viscosity, while B2/1 was molded at 350 °C in order to visualize the formation of the liquid crystalline phase. Polarized optical microscopy (POM) of B2/1 demonstrated mesophase formation (Figure 5), but due to the high melt viscosity it was difficult to obtain a sufficiently thin layer of the microslide.

In general, it can be said that the binary copolyesters of VA and HBCA do not have acceptable characteristics; these polymers have a softening point that is too high, close to the beginning of the decomposition of vanillic acid units. At the same time, despite the ability of polyesters to form a mesophase at a high VA content, such melts are unacceptably viscous for effective processing.

To reduce the softening point, we synthesized ternary copolyesters containing HBA units in addition to VA and HBCA.

### 3.2. Ternary Copolyesters

Wilsens et al. applied thin-film polycondensation with small amounts of monomers to study the composition of potential melt-processable copolyesters [37,43]. In the present study, as well as in our previous work [44], we applied small-scale (~1 g of monomers) polycondensation in a flask in an inert atmosphere, in the 240–320 °C range. This allowed more thorough visual control of the polycondensation process and the viscosity evolution. When the viscosity increased, the polycondensation was stopped and the product was evacuated from the flask. The results are presented in Table 2. Much attention was paid to the composition with reduced molar content of ABCA. The molar ratio of comonomer was chosen with consideration of our previous works: (1) The moiety of HBCA should be reduced as much as possible, because HBCA is a more expensive and fuel-based comonomer. However, using less than 20 mol% HBCA resulted in an intractable copolyester, while more than 33.3 mol% is not feasible. Thus, we used 33.3% and 26% HBCA. (2) Generally, the lowest melting of copolyesters was observed when the molar ratio of comonomers was roughly equal. Additionally, for the purpose of comparison with our previous work, the ternary copolyester T68/21/11 was prepared from AVA, ABCA, and 3-acetoxybenzoic acid (3ABA), which is similar to the previously described ternary copolyester composed of HBA (70%), HBCA (20%), and 3HBA (10%), which melted at 315 °C [38]. 

The copolyesters T1/1/1, T2/2/1, and T37/37/26 softened at ~300 °C. The preparation of T68/21/21 failed because, after 15 min of polycondensation of the monomers at 280 °C, the reaction mixture solidified, and increasing the temperature to 320 °C had no effect. This was unexpected, since the copolyester of HBA (70%), HBCA (20%), and 3HBA (10%) was melt-processable. Typically, the introduction of side substituents reduces the melting or softening point of the resulting copolyester. Although the VA unit is structurally an HBA unit with a bulky substituent, the HBA and VA units play different roles in thermal properties.

The ternary VA-HBA-HBCA copolymers are also non-crystalline. At the same time, they have noticeably lower glass transition temperatures (~117 °C) even compared to the binary VA-HBCA copolyesters (~135 °C) and especially compared to the 3HBA-3HBCA copolyesters (4′-hydroxybiphenyl-3-carboxylic), for which *T*_g_ can reach 190 °C [44]. As has been repeatedly shown, including in our early works [19,38], in polyesters based on aromatic hydroxy acids, a comonomer with biphenyl units—in this case, HBCA—makes a large contribution to heat resistance. The ratio of biphenyl and monophenyl comonomer units in the binary VA-HBCA copolymers is higher than in the ternary VA-HBA-HBCA copolymers, which leads to a decrease in *T*_g_ values.

As in the case of binary copolyesters, terpolymers also show a stepwise thermal decomposition pattern (Figure 6) with two weight loss maxima: around 410–430 °C and 520–530 °C. In addition, ternary copolyesters have noticeably lower temperatures for the onset of weight loss (300–330 °C) and 5% weight loss (390–410 °C). This is probably due to a decrease in the HBCA/VA ratio, since in the series T2/2/1, T37/37/26, and T1/1/1 an increase in thermal stability was noted with the increase in the content of biphenyl HBCA units. However, the weight of the residue after decomposition is higher for ternary copolyesters, which can be explained by the presence of a large portion of HBA units, which themselves have high thermal stability.

Next, a series of high-molecular-weight ternary copolyesters were obtained using melt polycondensation at 240–320 °C. A traditional method was applied, using a 100-mL three-neck round-bottomed flask equipped with an overhead glass stirrer, argon inlet, and vacuum outlet at a scale of 15–25 g. The temperature was gradually increased from 240 °C to 320 °C. The final stage was carried out at a temperature of 320 °C under vacuum. The polycondensation was stopped when the desired viscosity was reached. The products prepared are listed in Table 3. The polymers TM1/1/1 and TM1/2/1 did not dissolve in pentafluorophenol at 80 °C, where only swelling was observed.

During the melt polycondensation, darkening of the reaction mixture was observed at temperatures above 280 °C. To avoid destructive processes, a combined method was used. The first stage was carried out in the melt at a temperature of 240–280 °C in an argon atmosphere. Then, the prepolymers were crushed and polymerized in a solid state at 250–260 °C under vacuum or under an argon atmosphere for 8–16 h (Table 4). Despite the decrease in polycondensation temperature, colored products were still obtained. The copolyester TS1/3/1, with a lower VA content, was less colored compared to the others. Thus, the color change was caused by the presence of VA units in the copolyesters. All copolyesters listed in Table 3 and Table 4 were relatively ductile, and their logarithmic viscosity values were high. Some of the copolyesters were insoluble in pentafluorophenol due to their high molecular weight. For this reason, IR spectroscopy was also used to confirm the structure of the copolyesters.

The FTIR spectra of the ternary copolyesters are presented in Figure 7, along with all of the characteristic absorption bands as the spectra of the binary copolymers. The C=O vibration band (*ν*_C=O_) of the ester groups was shifted to 1728 cm^−1^. Stretching vibrations of the C–O bonds (*ν*_C-O_) in VA units appeared at 1283 cm^−1^ and 1028 cm^−1^ in the form of a shoulder due to the decrease in the number of VA units. Benzene ring stretching vibrations (*ν*_C=C_) and bending vibrations were present in the same regions as in the binary copolyesters: 1600, 1500, 763, 749, 713, and 694 cm^−1^.

X-ray diffraction patterns were obtained for a number of ternary polyesters with different comonomer ratios in the 2*θ* range from 5 to 60° (Figure 8). All of the ternary copolyesters showed no sharp reflections of crystalline phases, only a broad halo with a maximum of ~20°, indicating their low degree of crystallinity. All three comonomers (VA, HBA, HBCA) are structurally similar and have similar XRD patterns. The peak at 20° is characteristic of structural fragments of both HBCA [19,45] and HBA [46]. At the same time, a sharper peak is usually observed for biphenyl fragments of HBCA, and a broader one for HBA. Thus, with an increase in the HBCA/HBA ratio from TS1/3/1 to TS37/37/26, a narrowing of the reflex at 20° was noted. According to [34], for structural fragments of vanillic acid, peaks are observed in the region of 15–20° and at 25°, but when the ratio of VA to other comonomers is equalized, all peaks disappear. This explains the absence of additional peaks in the XRD patterns for terpolymers, especially considering their low contents of VA compared to other comonomers.

Terpolymers obtained by means of melt polycondensation and using a solid-state process demonstrated similar thermal characteristics. They had a relatively low glass transition temperature, in the region of 110–120 °C (Table 3 and Table 4), and a stepwise behavior of thermal decomposition (Figure 9 and Figure 10). For all samples, the typical temperature for the onset of mass loss was in the range of 330–350 °C, the temperature of 5% mass loss in the range of 390–410 °C, while the weight residue at 1000 °C was ~40%.

The ternary copolyesters with the same composition, but obtained using different methods (in solution, in the melt, and in solid phase), evidently possessed different molecular weights; no significant differences in thermal characteristics were observed. Table 5 shows the results for different VA/HBA/HBCA copolyester samples at a ratio of 1/1/1; no significant differences were found in the thermal stability of copolyesters subjected to solid-state polycondensation for different times (8 and 16 h). This was not unexpected; for example, the authors of [47,48] reported that for similar copolyesters, in a wide range of molecular weights, the thermal characteristics differed insignificantly.

In general, the introduction of VA into the macromolecular backbone drastically reduces the thermal stability. TGA showed that the decomposition started at 330 °C, so visual observations were conducted at 280 °C. Therefore, VA copolymers should preferably also contain aliphatic comonomers to reduce the processing temperature to below 280 °C.

The existence of a liquid crystalline phase of VA/HBA/HBCA ternary copolyesters with various comonomer compositions—1/1/1, 1/2/1, and 1/3/1 (Figure 11)—was confirmed via polarization optical microscopy (POM). Generally, TMCPs exhibit a schlieren texture typical of nematic melts. Similar results were obtained for copolyesters based on hydroxy acids and aromatic diols/diacids of various structures [19,43,49,50].

The ternary copolyesters did not demonstrate isotropization up to 375 °C, which was the apparatus limit and higher than the decomposition temperature. After cooling, liquid crystalline (LC) melts kept the texture; thus, the formation of LC glasses was observed. As an example, Figure 12 shows micrographs obtained by heating copolyester T1/1/1 from a temperature below the glass transition to the temperature of mesophase formation throughout the entire volume of the melt. It can be observed that when the glass transition temperature was exceeded and as the viscosity of the melt decreased, the formation, growth, and coalescence of LC-phase droplets occurred.

The mechanical characteristics of the ternary polyesters (tensile strength, relative elongation at break, and elasticity modulus) were measured by means of tensile testing of film samples. The thin-film copolyesters (thickness 100 μm) were molded by means of hot pressing at 300 °C. 

Tests were carried out for polyesters TM1/1/1, TM1/2/1, TS37/37/26, and TS1/3/1. Unfortunately, it was not possible to achieve high strength characteristics for these copolymers. The average values of their mechanical characteristics were as follows: tensile strength of 39 MPa, elastic modulus of 125 MPa, and relative elongation at break of 4%. These characteristics are close to those for semi-aromatic copolyesters of polyethylene terephthalate and HBCA [19], but they are much lower than the typical values for fully aromatic thermotropic polymers. We assume that this problem is associated with the fairly high softening temperature of the copolyesters, which lies close to the region of thermal degradation. In this regard, VA should be used for copolyesters with moderate thermal stability, and additional comonomers reducing the processing point should be included into the macromolecular backbone of the copolyesters.

Wilsens et al. used VA as a comonomer for TMCPs at molar ratios of 10 and 16% [13,25,51,52]. On the other hand, the ternary copolyester of HBA/HNA/VA with a 5% molar content of VA demonstrated a reduced melting point, increased glass transition temperature, good thermal stability, and excellent spinnability [53]. In combination with the results obtained in our research, it can be stated that vanillic acid can be used as an additive comonomer for the synthesis of TMCPs with contents not exceeding 10–20 mol%, and not being considered as the main comonomer. Higher amounts of VA in TMCPs result in decreased thermal stability and increased melt viscosity due to the presence of the bulky –OCH_3_ hindering chain mobility.

## 4. Conclusions

(1) In this research, we attempted to obtain fully aromatic polyesters using bioavailable vanillic acid as a comonomer. Several series of new polymers were synthesized: binary copolyesters of vanillic (VA) and hydroxybiphenylcarboxylic (HBCA) acids, as well as ternary copolyesters additionally containing hydroxybenzoic acid (HBA) and obtained in three different ways (in solution, in melt, and in solid state). The main focus was on varying the VA/HBA ratio, while the HBCA content was 26 or 33 mol%.

(2) The synthesized copolyesters had high molecular weights. Their structure was confirmed via FTIR and XRD methods; the data obtained are in full agreement with the literature data and our previous studies of polymers of this type.

(3) Using the method of simultaneous thermal analysis, the influence of the comonomer composition (i.e., the VA/HBA/HBCA ratio) on the thermal characteristics of the resulting copolyesters was shown. The use of vanillic acid led to the preparation of copolyesters with a relatively high glass transition temperature (~130 °C). At the same time, the heat resistance of binary VA-HBCA copolymers was higher than that of ternary VA-HBA-HBCA copolymers, which was explained by the decreased biphenyl moiety in ternary copolyesters in comparison with that of binary copolyesters. The onset of weight loss of ternary copolymers occurred at 330–350 °C, while the temperature of 5% mass loss was in the range of 390–410 °C. It was observed that all binary and ternary copolyesters of vanillic acid demonstrated two-stage thermal destruction: the decomposition started with VA units at 420–480 °C, and then the decomposition of more heat-resistant units occurred above 520 °C. An increase in the proportion of VA leads to an increase in the integral thermal effect of the decomposition below 500 °C.

(4) The possibility of the transition of polyesters to the liquid crystalline state was studied using polarization optical microscopy. It has been shown that, when heated above the glass transition points, the copolyesters are thermotropic and exhibit a typical nematic type of liquid crystalline state in the melt. Due to the increase in melt mobility, with increasing temperature, the formation of liquid crystal-phase droplets and their coalescence accelerates. Ternary VA-HBA-HBCA copolymers form LC melts more readily due to their lower melting temperature, in contrast to binary VA-HBCA copolymers. The isotropization temperatures were higher than the temperature of thermal destruction for all copolyesters.

(5) The mechanical characteristics of the copolyesters were similar to those of semi-aromatic copolyesters, but they were much lower than those of fully aromatic thermotropic polymers. We assume that this problem is associated with the fairly high softening temperature of such copolyesters, which lies close to the region where thermal degradation begins. In this regard, VA should be used for copolyesters with moderate thermal stability, and additional comonomers, reducing the processing point, should be included into the macromolecular backbone of the copolyesters. 

(6) Summarizing the data, vanillic acid can be used as an additive comonomer for the synthesis of TMCPs at contents not exceeding 20 mol% (more preferably not exceeding 10 mol%) and not being considered as the main comonomer. Higher amounts of VA in TMCPs result in decreased thermal stability and increased melt viscosity.

## Figures and Tables

**Figure 1 polymers-16-01501-f001:**
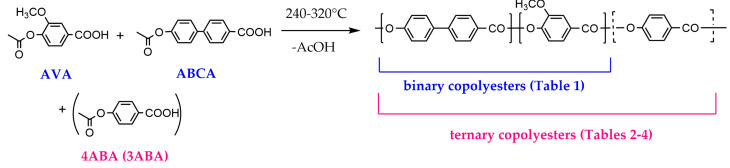
Binary and ternary copolyester synthesis route.

**Figure 2 polymers-16-01501-f002:**
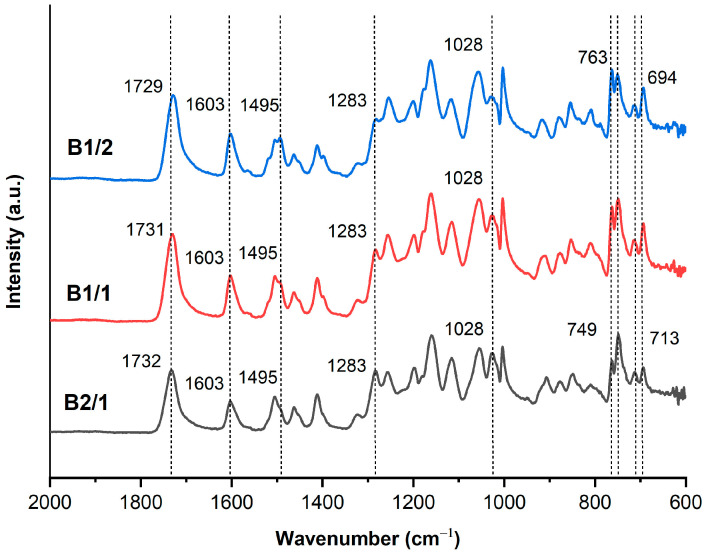
FTIR-ATR spectra of binary copolyesters based on VA and HBCA (B2/1, B1/2, and B2/1).

**Figure 3 polymers-16-01501-f003:**
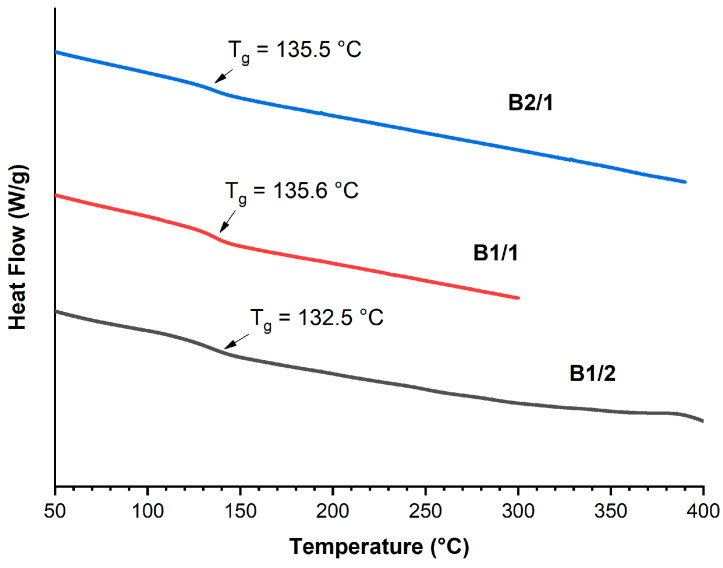
DSC thermograms of binary copolyesters based on VA and HBCA (B2/1, B1/2, and B2/1) in the second heating cycle, at a rate of 10 °C/min, in an inert atmosphere.

**Figure 4 polymers-16-01501-f004:**
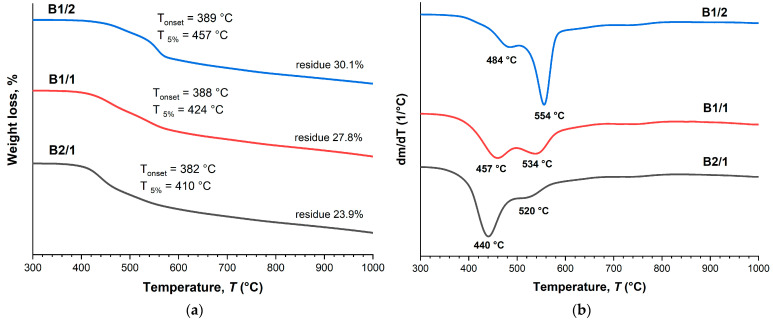
TGA (**a**) and DTG (**b**) curves for binary polyesters based on VA and HBCA (B2/1, B1/2, and B2/1) in an inert atmosphere, at a heating rate of 10 °C/min.

**Figure 5 polymers-16-01501-f005:**
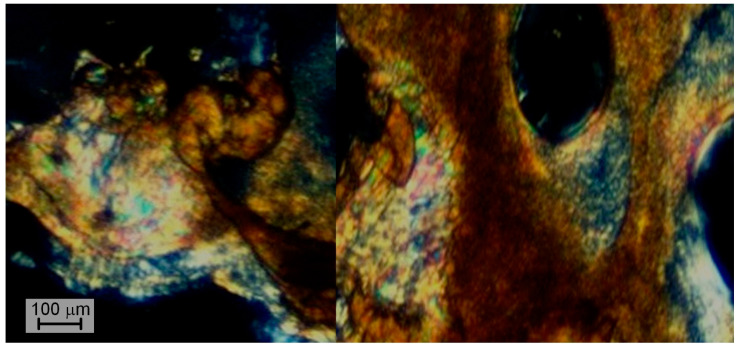
Polarized optical microscopy of binary copolyester B2/1 melted at 350 °C.

**Figure 6 polymers-16-01501-f006:**
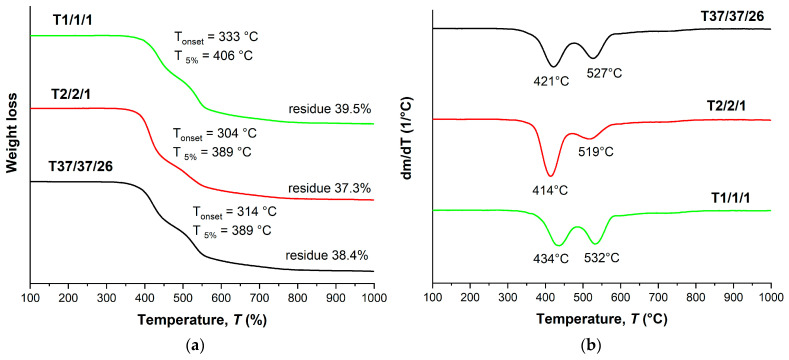
TGA (**a**) and DTG (**b**) curves for ternary polyesters based on VA, HBA, and HBCA, prepared by means of small-scale melt polycondensation; analysis performed in an inert atmosphere at a heating rate of 10 °C/min.

**Figure 7 polymers-16-01501-f007:**
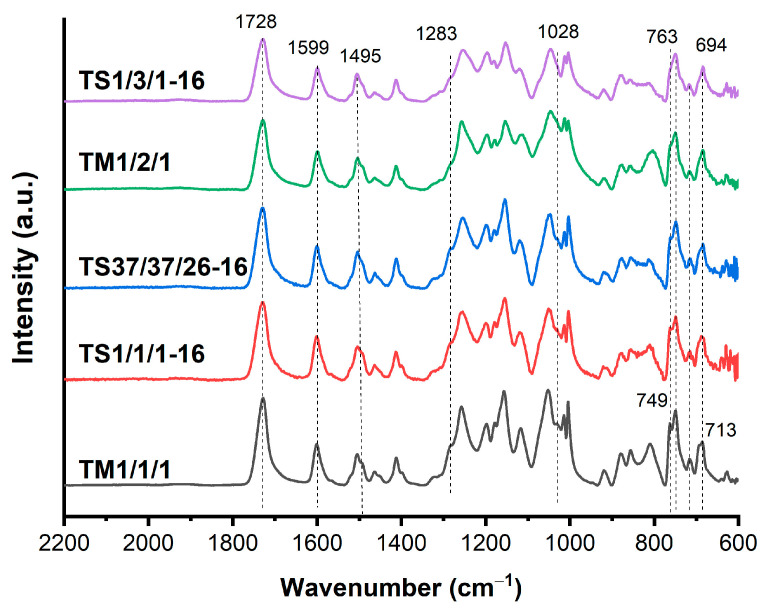
FTIR-ATR spectra of ternary copolyesters of VA, HBA, and HBCA, prepared by means of melt polycondensation and solid-state polycondensation.

**Figure 8 polymers-16-01501-f008:**
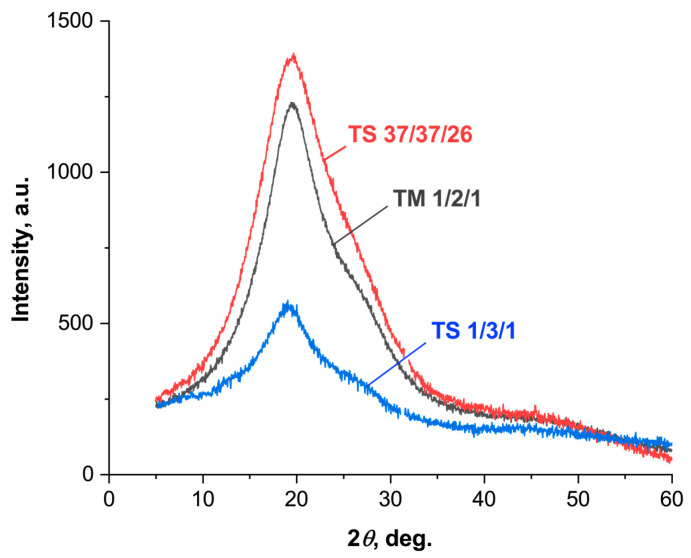
XRD patterns of ternary copolyesters of VA, HBA, and HBCA.

**Figure 9 polymers-16-01501-f009:**
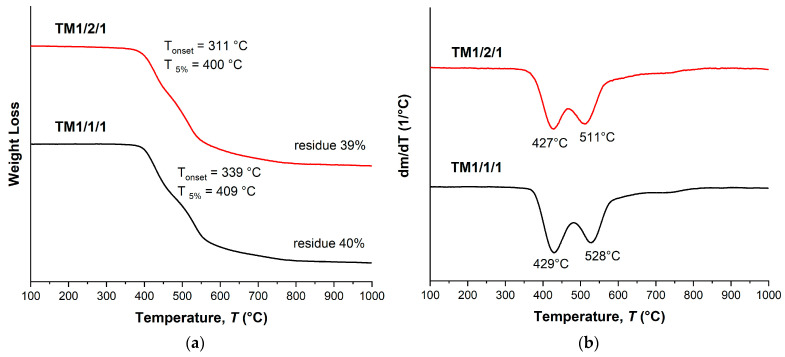
TGA (**a**) and DTG (**b**) curves for ternary copolyesters prepared by means of melt polycondensation of AVA, ABA, and ABCA; analysis performed in an inert atmosphere at a heating rate of 10 °C/min.

**Figure 10 polymers-16-01501-f010:**
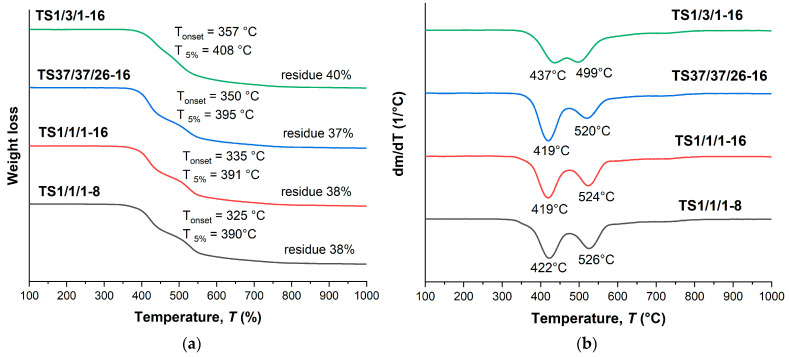
TGA (**a**) and DTG (**b**) curves for ternary copolyesters based on AVA, ABA, and ABCA, prepared by means of solid-state polycondensation of prepolymers; analysis performed in an inert atmosphere at a heating rate of 10 °C/min.

**Figure 11 polymers-16-01501-f011:**
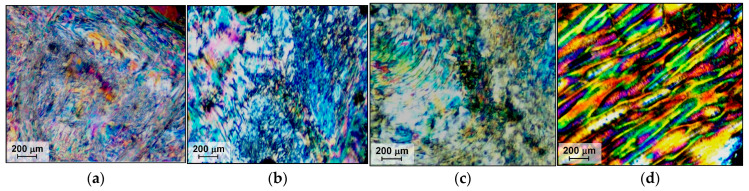
Polarized optical microscopy of ternary copolyesters: T1/1/1 at 300 °C (**a**); T37/37/26 at 300 °C (**b**); TM1/2/1 at 340 °C (**c**); TS1/3/1 at 300 °C (**d**).

**Figure 12 polymers-16-01501-f012:**
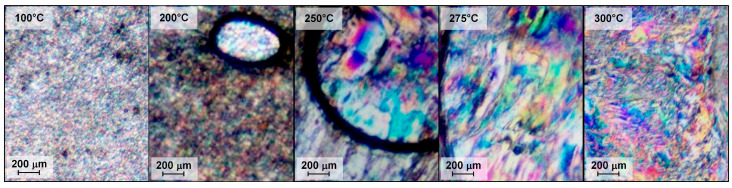
Polarized optical microscopy of ternary copolyester T1/1/1 during its heating from 100 to 300 °C at a rate of 10 °C/min.

**Table 1 polymers-16-01501-t001:** Comonomer ratio, glass point (*T*g), melting point (*T*m), temperature of weight loss onset (*T*on), and temperature of 5% weight loss (*T*_5%_) of binary copolyesters based on VA and HBCA.

Copolyester	Molar Ratio (AVA/ABCA)	*T*_g_, °C	*T*_m_, °C	*T*_on_, °C	*T*_5%_, °C
B 1/2	1:2	135.5	-	389	457
B 1/1	1:1	135.6	-	388	424
B 2/1	2:1	132.5	-	382	410

**Table 2 polymers-16-01501-t002:** Comonomer ratio, glass point (*T*g), melting point (*T*m), temperature of weight loss onset (*T*on), and temperature of 5% weight loss (*T*_5%_) of ternary copolyesters prepared by means of small-scale melt polycondensation of AVA, ABA, and ABCA.

Copolyester	Molar Ratio (AVA/ABA/ABCA)	*T*_g_, °C	*T*_m_, °C	*T*_on_, °C	*T*_5%_, °C
T 1/1/1	1:1:1	117.5	-	332.7	405.9
T 2/2/1	2:2:1	116.4	-	304.4	389.2
T 37/37/26	37:37:26	116.8	-	313.8	388.9
T 68/21/21 ^1^	68:21:11	-	-	-	-

^1^ 3-Acetoxybenzoic acid was used instead of 4-acetoxybenzoic acid.

**Table 3 polymers-16-01501-t003:** Comonomer ratio, glass point (*T*g), melting point (*T*m), temperature of weight loss onset (*T*on), and temperature of 5% weight loss (*T*_5%_) of ternary copolyesters prepared by means of melt polycondensation of AVA, ABA, and ABCA.

Copolyester	Molar Ratio (AVA/ABA/ABCA)	*T*_g_, °C	*T*_m_, °C	*T*_on_, °C	*T*_5%_, °C
TM 1/1/1	1:1:1	120.8	-	339.1	408.7
TM 1/2/1	1:2:1	117.0	-	310.5	399.7

**Table 4 polymers-16-01501-t004:** Comonomer ratio, glass point (*T*g), melting point (*T*m), temperature of weight loss onset (*T*on), and temperature of 5% weight loss (*T*_5%_) of ternary copolyesters based on AVA, ABA, and ABCA, prepared by means of solid-state polycondensation of prepolymers. The SSP conditions are also indicated.

Copolyester	Molar Ratio (AVA/ABA/ABCA)	SSP Condition: Temperature, °C (Time, h)	[*η*]	*T*_g_, °C	*T*_m_, °C	*T*_on_, °C	*T*_5%_, °C
TS 1/1/1-8	1:1:1	250 (8)	6.3	114.0	-	325.1	389.8
TS 1/1/1-16		250 (8) + 255 (8)	8.7	116.8	-	334.9	391.4
TS 37/37/26-8	37:37:26	250 (8)	13.8	113.1	-	*n/a*	*n/a*
TS 37/37/26-16		250 (8) + 255 (8)	*insol.*	119.6	-	349.7	394.9
TS 1/3/1-8	1:3:1	250–260 (8)	*insol.*	106.7	-	*n/a*	*n/a*
TS 1/3/1-16		260 (16)	*insol.*	108.1	-	357.3	408.2

**Table 5 polymers-16-01501-t005:** Comparison of the thermal characteristics of ternary copolyesters (VA-HBA-HBCA) with the same comonomer composition, obtained via different methods.

Copolyester	Molar Ratio (AVA/ABA/ABCA)	[*η*]	*T*_g_, °C	*T*_on_, °C	*T*_5%_, °C	Residue at 1000 °C
T 1/1/1	1:1:1	1.2	118	333	406	40
TM 1/1/1		*insol.*	121	339	409	40
TS 1/1/1-8		6.3	114	325	390	38
TS 1/1/1-16		8.7	117	335	391	38

## Data Availability

The raw data supporting the conclusions of this article will be made available by the authors on request.

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
