# Peer review of "Fully Aromatic Thermotropic Copolyesters Based on Vanillic, Hydroxybenzoic, and Hydroxybiphenylcarboxylic Acids"

_polymers, 2024, doi:10.3390/polym16111501_

Round 1
Reviewer 1 Report
Comments and Suggestions for Authors
This manuscript aims to confirms that vanillic acid is a mesogenic monomer suitable for the synthesis of thermotropic fully aromatic and semi-aromatic copolyester, the high values of logarithmic intrinsic viscosities and insolubility of several samples proved its high molecular weights. The experiment has certain feasibility, but minor modifications are still needed before publication.
1. In the introduction, when mentioning TMCP fiber and aramid fiber, aramid fiber was not mentioned earlier, and the comparison here is a bit abrupt. Please make further adjustments. 2. In the introduction, emphasis should be placed on vanillic acid, 2-hydroxybenzoic acid, and hydroxybenzoic acid. Please provide a detailed description of these three substances. 3. Please include multiple datasets with varying molar ratios in Table 2 of Section 3.2 to enhance the persuasiveness of your selected molar ratio. 4. Please reorganize the conclusion section, expand the description, and divide it into several points to make it more organized. Comments on the Quality of English LanguageExtensive editing of English language required.
Author Response
Dear Reviewer!
We thank you for your good evaluation of our work and for your time devoted to reviewing.
Below are our responses to your comments.
1) In the introduction, when mentioning TMCP fiber and aramid fiber, aramid fiber was not mentioned earlier, and the comparison here is a bit abrupt. Please make further adjustments.
At your suggestion, fragments were added to the Introduction (lines 29-31, 37-41), where a more detailed comparison of materials obtained from thermotropic and lyotropic polymers was carried out. The comparison was also supplemented with two new references (no. 15 and 16).
TMCPs are capable of forming highly ordered liquid crystalline melts and do not require solvents during processing, unlike materials produced from lyotropic solutions such as aramid fibers. <…> The mechanical properties of TMCP fibers are comparable to those of aramid fibers [6,15,16]. For instance, the tensile strength of Vectran HS fibers was found to be ~3 GPa [15]. An early patent showed that the tensile strength of Kevlar fibers was only slightly higher than that of TMCP fibers (36.9 g per denier vs. 30.8 g per denier) [16]. It should be noted that the moisture uptake of TMCPs is significantly lower than that of aramid fibers [3].
2) In the introduction, emphasis should be placed on vanillic acid, 2-hydroxybenzoic acid, and hydroxybenzoic acid. Please provide a detailed description of these three substances.
At your suggestion, fragments were added to the Introduction (lines 66-67, 70-77), where emphasis was placed on the reasons for choosing vanillic acid and 4-hydroxybenzoic acids as the main comonomers for the synthesis of polyesters.
The great interest in both the academic and industrial fields over the past decades has been the use of monomers from bio-based renewable sources. Lignin and cellulose are the most abundant renewable feedstocks. The most promising lignin-based polyester monomers are phenolic acids: vanillic, syringic, 4-hydroxybenzoic, floretic (3-(4-hydroxyphenyl)propionic), 4-hydroxycinnamic, ferulic (3-methoxy-4-hydroxycinnamic), dihydroferulic, etc. [20–28]. 4-Hydroxybenzoic and vanillic acids are aromatic self-condensable AB-type monomers. It should be noted that 4-hydroxybenzoic acid can be produced both from conventional oil-based sources and from renewable feedstocks, while the most promising source of VA is lignin. Structurally, vanillic acid is similar to 4-hydroxybenzoic acid, excluding the presence of the bulky side -OCH3 group, possessing mesogenic properties, and suitable as a comonomer for the synthesis of TMCPs. A role of the bulky side group of vanillic acid is that it disrupts the molecular order in copolymers, thereby reducing their melting or softening point.
3) Please include multiple datasets with varying molar ratios in Table 2 of Section 3.2 to enhance the persuasiveness of your selected molar ratio.
A fragment was added on lines 305-312, where the choice of comonomer ratios for the synthesis of ternary copolyesters was explained:
The molar ratio of comonomer was chosen with consideration of our previous works: (1) The moiety of HBCA should be reduced as much as possible, because HBCA is a more expensive and fuel-based comonomer. However, using less than 20 mol% HBCA resulted in an intractable copolyester, while more than 33.3 mol% is not feasible. Thus, we used 33.3% and 26% HBCA. (2) Generally, the lowest melting of copolyesters was observed when the molar ratio of comonomers was roughly equal. Additionally, for the purpose of comparison with our previous work, the ternary copolyester T68/21/11 was prepared from AVA, ABCA, and 3-acetoxybenzoic acid (3ABA), which is similar to the previously described ternary copolyester composed of HBA (70%), HBCA (20 %), and 3HBA (10%), which was melted at 315 °C.
4) Please reorganize the conclusion section, expand the description, and divide it into several points to make it more organized.
The Conclusion section has been updated. At your suggestion, the section was divided into paragraphs and supplemented with additional explanations on lines 485-492, 501-504, 506-511.
5) Extensive editing of English language required.
Proof reading of the manuscript was performed via MDPI’s Author Services.
Sincerely, the team of authors.
Reviewer 2 Report
Comments and Suggestions for Authors
The manuscript by Kulichikhin et al. reports on their synthesis and physico-chemical characterization of two new families of copolymers of vanillic and 8-hydroxybiphenylcarboxylic acid (in the case of binary copolymers) and hydroxybenzoic acid (ternary copolymers). The primary focus of the manuscript is effect of presence and fraction of hydroxybenzoic acid (and therefore comparison of binary and ternary copolymers) as well as vanillic acid on thermomechanical properties and processing of the reported copolymers. The work is original, the references are relevant and up to date, and the introduction gives a clear insight into the problem. The results are clearly reported and adequately discussed. The paper may be of interest to a broad audience of Polymers.
The drawback of the manuscript is poor English. Some phrases sound ridiculous and awkward. I may strongly recommend the authors to get help of an expert in technical/scientific English.
To summarize, I recommend extensive editing of English language, upon which the manuscript may be published.
Comments on the Quality of English LanguageExtensive editing of English is required.
Author Response
Dear Reviewer!
We thank you for your high evaluation of our work and for your time devoted to reviewing.
Below are our responses to your comment.
- The drawback of the manuscript is poor English. Some phrases sound ridiculous and awkward. I may strongly recommend the authors to get help of an expert in technical/scientific English.
Proof reading of the manuscript was performed via MDPI’s Author Services.
Sincerely, the team of authors.